# Functional Ego States, Behavior Patterns, and Social Interaction of Bulgarian Adolescents and Their Parents

Sezgin Bekir * and Ergyul Tair

Institute for Population and Human Studies, Bulgarian Academy of Sciences, 1000 Sofia, Bulgaria; egyul_tair@yahoo.com
* Correspondence: sezgin1974@abv.bg; Tel.: +359-886-318-922

**Abstract:** Adolescence is a dynamic period of transition, and interaction with parents is important for successfully passing through it. This article presents results from a study of three transactional analysis theory personality constructs of adolescents and their parents—functional ego states, life positions, and stroke economy. The sample included 215 students from 12 to 16 years old and 215 parents from 30 to 55 years old. The results show significant differences between the ego states Adult, Adapted Child, Nurturing Parent and Normative Parent of the adolescents and their parents, which are more expressed in the parents. It was found that the life position, "I am OK—you are OK" is more strongly expressed in parents, while the life position, "I am not OK—you are not OK" is leading in adolescents. In their interactions with others, adolescents express modesty, self-doubt, and underestimation of their own success, while parents are more assertive, confident, and resilient in their behavior. The obtained results can be used in different programs to improve social interaction and increase the effectiveness of adolescent-parent relationships.

**Keywords:** personality; life position; stroke economy; family environment; transactional analysis



## 1. Introduction

On the basis that psychoemotional development and the formation of the core structure of personality are early life experiences, with family and social environment playing an important role in this process (e.g., [1–5]). The role of the family in social development and parent-adolescent relationships has a constant presence in the literature on adolescence [6–8]. From a very young age, children become accustomed to certain patterns of interaction with their parents and/or other caring adults [1,9]. Furthermore, in their interactions with other people, an adult tends to repeat the same patterns of behavior that they learned and adopted as a child. In this sense, relationships from earlier stages of development with parents and/or other family members influence both the formation of the adolescent's own identity and his or her relationships with the social environment as a whole.

A young person's personality is formed and developed as a result of interaction with significant others in the process of socialization [10]. Socialization, which primarily takes place in the family environment, aims to shape adolescents' appropriate school behavior (e.g., [11]), effective peer relationships (e.g., [12]), positive family relationships, and the ability to adapt behavior to the demands of society (e.g., [13]). However, relationships with parents in adolescence differ from attachment in early childhood on behavioral, cognitive and emotional levels [14]. The intense experiences and personal development during adolescence require a more attentive and different parental approach to the young person, such as supporting the pursuit of autonomy, providing opportunities to explore and experience different things, and providing a sense of a secure family environment [15]. Interpersonal relationships have an important role in the accumulation of experience, which subsequently enables the regulation of adolescents' behavior and emotional responses. In this paper, we present the results of a study of the personality structure (functional ego states) and behavior patterns, measured by life positions and strokes economy of 12–16-year-old adolescents

and their parents. We did not find in the literature any quantitative studies that explore the relationships between ego states, life positions and stroke economy in adolescents and their parents, except our mentioned studies in Bulgarian conditions. This may be due on the one hand to the limited number of quantitative methods with good psychometric characteristics for measuring different constructs from the theory of transactional analysis [16,17], or to the still insufficient development of quantitative empirical research in the field of transactional analysis [17,18]. We consider it important to explore the different personality characteristics and behavior patterns of both adolescents and parents in order to support effective parenting and ensure a smooth transition of adolescents from childhood to adulthood and autonomous functioning.

### 1.1. Models for Representing Personality Structure and Behavior in Transactional Analysis

Transactional analysis (TA) originated as a theory of personality and a psychotherapeutic approach. Today, TA finds wide application as both a theory of communication and a theory of child development [1,18]. In the present study, we aimed to examine the personality characteristics of adolescents and their parents through three main constructs proposed in TA, namely: functional ego states, life positions, and strokes. Ego states represent a coherent system of thoughts, feelings and behaviors and are a core theoretical construct in TA. Berne introduced the structure of personality metaphorically represented as consisting of three ego states, Parent, Adult, and Child, which are manifested through consistent behavioral patterns [9,19,20]. The Parent ego state includes the thoughts, feelings, and behaviors internalized in infancy by parental figures. The primary function of the Adult ego state involves the individual's ability to recognize and distinguish between internal experiences, external reality, and reality testing. The Child ego state is a structure related to emotions, instincts, needs and stores within itself all the past experiences of the individual.

### 1.2. Functional Ego States Model

A manifestation of the three ego states in social interaction and their expression in specific behaviors is analyzed through a functional model, which includes five functional ego states: Adult, Normative Parent, Nurturing Parent, Adapted Child and Natural Child [1,18,19]. The adult is associated with the rational, analytical and logical aspects in personal behavior. The Nurturing Parent is manifested through caring, supporting and encouraging others, while the Normative Parent is associated through expressing criticality, setting boundaries and adhering to rules and norms. Emotional, impulsive, and spontaneous behavior are functions of the Natural Child, while the Adapted Child includes the tendency to obey, adapt, and conform to the demands of others. Although in age terms the formation of ego states is not characterized by simultaneity, it is assumed that by the age of 12 the Parent, Adult and Child are established [20,21]. All subsequent experiences in a person's life after this period do not bring new content to the Parent and Child, but rather, in the period of adolescence, already existing records are reinforced or reintegrated [18,20]. At the same time, the Adult ego state continues to develop throughout the person's life, being activated in any situation where a decision is to be made here and now [9,21]. Research on correlation between ego states and self-control in students disclose that students with high parent ego states make decisions using reformative self-control, students with high adult ego states make decisions using reformative and redressive self-control, and students with high child ego states make decisions using experiential self-control [22]. The results of our and other available studies show that the Nurturing Parent ego state has a strong positive relationship with the personality trait agreeableness, the Adult ego state has a positive relationship with the personality trait conscientiousness and the Natural Child ego state has a positive relationship with the personality trait extraversion and career satisfaction [23–25].

### 1.3. Life Positions

In TA, the attitude towards the self and the outside world formed in early childhood as a result of messages received from parents is described by the life positions [1,9,26]. Social relationships and an individual's choice to exhibit different patterns of behavior depending on the social context are influenced by life positions. Life position is made up of relatively stable personality characteristics, and four basic positions have been described in the literature: "I am OK—you are OK", "I am not OK—you are OK", "I am OK—you are not OK" and "I am not OK—you are not OK" [1]. The position "I am OK—you are OK" is associated with sociability, empathy and a well-built image of oneself and others. Life position "I am OK—you are not OK" is associated with emphasizing one's own importance and devaluing others, while the position "I am not OK—you are OK" involves experiencing feelings of inferiority and difficulty adapting to one's environment. Life position "I am not OK—you are not OK" refers to a constructed negative view of self and others. Results from our research on satisfaction with career success show that the life position "I am OK—you are OK" is dominant in parents of adolescents, which is characterized by the display of assertive behavior, following rules and setting boundaries in relationships [24].

### 1.4. Strokes and Stroke Economy

The strokes are the basic unit of social interaction in transactional analysis theory. Berne introduces the concept of "strokes", a form of social recognition or validation that people give and receive in their interactions [9]. Claude Steiner expanded on Berne's work by delving more deeply into the emotional aspects of transactional analysis [26]. His contributions often explored the ways in which positive and negative strokes and stroke economy impact emotional wellbeing. He helped further develop and popularize TA, contributing significantly to its body of literature. The exchange of strokes takes place through "transactions between those involved in the communication process and constitutes an important part of social relations" [9,19]. Receiving strokes, according to Berne, is one of the basic human needs and arises from the moment a child is born [1,9,18,26]. They can be positive and negative, unconditional and conditional, verbal and non-verbal. It is assumed that the free exchange of strokes at an early age is tightly controlled by parental figures, so that children develop a tendency to save giving or reject receiving strokes according to parental wishes [26]. Steiner defines this parsimony and form of parental control with the concept of stroke economy and describes five rules for their expression in behavior: "Don't accept strokes if you want them", "Don't ask for strokes when you need them", "Don't give strokes if you have them to give", "Don't reject strokes when you don't want them" and "Don't give yourself strokes". The stroke economy involves a relatively constant pattern of rejecting or saving identical messages in social interaction. In our research, for example, we found that for seventh-grade students, "Don't give yourself" and "Don't ask" were the leading styles, indicating a display of insecurity, passivity, and underestimation of one's own accomplishments in a school setting [27]. Higher expression of stroke economy in relationships is associated with displaying non-assertive and passive behaviors, while lower use of them in communication is a prerequisite for displaying assertive behaviors. Assertive behavior is assumed to involve the free giving and receiving of positive and negative strokes, whereas passive or non-assertive behavior involves the devaluation of one's own needs and over-adaptation to the environment [28].

### 1.5. Family Environment and Parent-Adolescent Relationships

A number of authors have emphasized the significant role of parents in the socialization process as well as the parent-adolescent relationship as an important factor in the adolescent's emotional and personal development process [5,7,29]. Parent-child relationships change over time, and these changes are particularly dynamic during adolescence as a result of the young person's quest for autonomy, search for identity, and frequent changes in behavior. Having parental support and encouragement increases adolescents' motivation as well as feelings of being accepted and valued [30]. For example, parents'

values, expectations, and supportive behaviors have been found to be positively associated with adolescents' self-esteem, need for achievement, developing competence, and striving for autonomy [31,32]. A number of parents' characteristics have significant effects on adolescents' behavior and social functioning [33]. For example, provision of resources, emotional warmth, and support from parents increases their influence on adolescents [34]. Children who are more liked by their peers and possess better social skills have been found to have friendlier and more emotionally and positively attuned mothers [35]. There is also evidence that children who are unaccepted and rejected by their peers have fathers who exhibit anger and other negative emotions more frequently [36]. At the same time, the expression of negative emotions by parents has a negative effect on the normal development and self-regulation of adolescents, which in turn is a prerequisite for the emergence and development of conflict and behavior problems outside the family environment [37].

The families provide an important social environment in which adolescents learn emotions and patterns of expressing their emotionality, and any change in this system also affects the parent-adolescent relationship [38]. The quality of the parent-child relationship may also result from adolescents' behavior, as their individual and personality characteristics determine different susceptibility to parental influence [39]. The effectiveness of the parent-adolescent relationship has a significant impact on adolescents' social life as well as on their physical and mental health [8,14,40]. At the same time, conflicts in the family environment, lack of support and difficulties in the parent-adolescent relationship are often identified as risk factors for physical and mental health impairment and the emergence of risky behaviors, including a tendency to alcohol and drug use [40,41].

In each family, a certain pattern of interaction and behavior develops, which is largely determined by the individual and personal characteristics of parents and their children [7,29,42]. However, the parent-adolescent relationship is not static and can change depending on the context in the short or long term in order to achieve better relationship performance [42]. In transactional analysis, it is assumed that the structure of personality and its functioning later as an adult is mainly determined by the interaction with parents in early childhood [1,9,18,26]. The ego states and life positions are formed in the period 0–12 years, and one of the growth tasks during adolescence is revisiting earlier development and experiences, which are readily available, and updating any patterns as necessary [20]. Our previous studies of the ego states and life positions of adults and adolescents in Bulgarian conditions show that the ego states Nurturing Parent, Adult and Natural Child and the life position "I am OK—you are OK" are the leading ones in both groups studied [43,44]. These results give us reason to assume that there is a stable structure of ego states and life positions as personality characteristics in adolescence [1,9,19,28]. From this point of view, it is important to examine the presented personal characteristics, as well as the stroke economy as a model of interaction between adolescents and their parents. This will allow the accumulation of new scientific facts about the constructs proposed in transactional analysis theory, as well as add to the knowledge about the complex adolescent-parent relationship in this often-challenging period of transition and change. Although it is well known that parents' ways of raising their children significantly affect their children's personality and behavior [22], this study has been designed because there is a lack of specific research on which ego states and life positions are associated with stroke economy in adolescents and their parents. In the present study, ego states, life positions and stroke economy have been taken into consideration together and answers to the following questions were sought:

1. Is there any difference in functional ego states, life positions, and strokes economy between the adolescents and their parents?
2. Do ego states and life position predict stroke economies?

The main objective of the present study is to examine functional ego states, life positions, stroke economy and to display patterns of social interaction in adolescents and their parents.

To realize the aim of the research, the following tasks are set:

1.　Establishing the expression of functional ego states, life positions, and stroke economy in adolescents and their parents.
2.　Establishing differences between ego states, life positions, and stroke economy in adolescents and their parents.
3.　Examining the impact of ego states and life positions on the stroke economy in the adolescents and parents studied.

On the basis of the theoretical review, we derive the following hypotheses:

1.　In the adolescents and parents studied, Nurturing Parent ego state, Adult ego state and the life position "I am OK—you are OK" will be expressed. We also assume that in adolescents "Don't give yourself" and "Don't ask" will be more expressed and in parents ""Don't ask" strokes economy" will be more expressed.
2.　We allow for significant differences in functional ego states, life positions, and strokes economy between the adolescents and parents studied.
3.　We expect the significant predictors of strokes economy in parents and adolescents to be the Normative Parent and the Adapted Child ego states, the life position "I am not OK—you are OK" and "I am not OK—You are not OK".

## 2. Methods

The sample. The study was conducted between June 2020 and October 2021 in the districts of Kardzhali, Razgrad and Targovishte as part of the implementation of activities under the project "Personality, education and professional self-realization for people from vulnerable groups in the country: scientific dimensions and practical approaches" [24,43,45]. Participants were a random sample of 7th-grade students from secondary schools in these three districts. The sample includes 215 adolescents from 12 to 16 years old (the mean age was 13.65 years and SD = 0.54) and 215 parents (mother or father) from age 30 to 55 years (the mean age was 41.44 years and SD = 5.76). The gender distribution of the sample is disproportionate, with adolescents dominated by boys (58.4%) and parents dominated by women (69.8%). Most of the families included in the survey were with two children (62.3%), 17.2% of the families were with one child, and 20.5% of the families were with three or more children. Data were collected in the schools, following permission from the Ministry of Education and Science of Bulgaria. All procedures performed in studies involving human participants were in accordance with ethical standards and informed consent was obtained from all individual participants included in the study. The schools and the classes were randomly selected, and adolescents' and parents' participation in the survey was voluntary and anonymous.

In the study, the following questionnaires were implemented:

(1)　Self-assessment questionnaire for the study of functional ego states [19], adapted for Bulgarian conditions [44]. The questionnaire contains 25 statements and scores are on a five-level Likert-type scale (from 1 = "It is not true in general for me" to 5 = "It is completely true for me"). The Normative Parent ego state is measured by items such as "I impose my opinion in solving problems", the Nurturing Parent is measured by items such as "I am dedicated to the needs of others" and the Adult ego state is measured by items such as "I consider things through before I act". The Adapted Child scale includes statements such as "I easily agree with the opinion of others" and the Natural Child ego state is measured by items such as "I take time for my own needs". Testing the reliability of the questionnaire using Cronbach's alpha shows that the reliability coefficient for adolescents it is 0.81 and for parents is 0.85, indicating an acceptable internal consistency of the items for the purposes of the study.

(2)　A self-report life position questionnaire developed by Bekir and Tair [24] with 24 items. Scores are on a five-level Likert-type scale (from 1 = "It is not true in general for me" to 5 = "It is completely true for me"). The life position "I'm OK—you are OK" is measured by adjectives such as "respectful", the position "I'm OK—you are not OK" includes adjectives such as "aggressive", the life position "I'm not OK—you are OK" is measured by statements like "yielding" and "I'm not OK—you are not OK" position

is measured by adjectives such as "dissatisfied". The resulting Cronbach's α reliability coefficient of the questionnaire for adolescents was 0.64 and for parents was 0.67, indicating acceptable psychometric properties and internal consistency of the items.

(3)   Self-report questionnaire for the study of stroke economy with 15 items [19], adapted for Bulgarian conditions [27]. Scores are on a five-level Likert-type scale (from 1 = "Totally disagree" to 5 = "Totally agree"). The "Don't Accept" stroke economy is measured by statements such as "I feel uncomfortable when others express their thoughts about me", the "Don't Give Yourself" is measured by statements such as "I do not reward myself for my success" and the "Don't Ask" stroke economy consists items such as "I don't seek compliment for something when other people have helped me to do it". The "Don't Give" stroke economy is measured with items such as "If you praise someone, he/she stops making effort" and the "Don't Reject" stroke economy is measured by items such as "Not accepting a compliment is a manifestation of lack of manners". Testing the reliability of the questionnaire using Cronbach's alpha indicates that the reliability coefficient for adolescents it is 0.58 and for parents is 0.76, indicating acceptable psychometric properties and internal consistency of the items.

*Statistical Analysis*

Means and standard deviations for each dimension were computed, and skewness and kurtosis normality indices were estimated to analyze data distributions. Differences in ego states, life positions and stroke economy between the adolescents and their parents were tested using a Student's independent-samples *t*-test [46]. The internal consistency of the psychological scales was estimated with Cronbach's alpha. Skewness and Kurtosis showed values are in the range ± 1, supporting normal distribution of data [46,47]. To analyze the role of ego states and life positions in stroke economy, a one-step regression analysis with the Enter method was calculated. In the regression model, the ego states and life positions were independent variables and dependent variables were stroke economy dimensions. SPSS V.13.0 [48] was used to calculate descriptive statistics, Student's independent-samples *t*-test and regression analysis. Tables and results shown on them are elaborated by the authors themselves.

## 3. Results

### 3.1. Descriptive Statistics of Ego States

Following the purposes and objectives of the study, we first present the results of the descriptive statistics conducted. The values of the descriptive statistics of the ego states presented in Table 1 show that for parents the leading ego states are Adult and Nurturing Parent, and the least manifested ego state is Normative Parent. For adolescents, the leading ego states are Nurturing Parent, Natural Child, and Adult, and again the least manifested ego state is Normative Parent. The results confirm the first hypothesis put forward in the part of dominance of the Nurturing Parent and Adult ego states. The results obtained are similar to the results of our other studies on the expression of functional ego states in adolescents and adults [43,44].

Ego States Differences between the Adolescents and Their Parents

Differences between ego states, also shown in Table 1, were tested using a Student's independent-samples *t*-test [46]. Significant differences were found between Adult, Adapted Child, Nurturing Parent and Normative Parent ego states of adolescents and their parents. No significant difference was found only in the Natural Child ego state. The results obtained confirm the second hypothesis put forward in the part of the existence of differences between the ego states of adolescents and their parents.

**Table 1.** Mean, standard deviation, normality indices, and differences in ego states between the adolescents and their parents.

| Dimensions | Respondents | N | Mean | SD | Skewness | Kurtosis | t-Test | p |
|---|---|---|---|---|---|---|---|---|
| Normative Parent | Parents | 215 | 15.21 | 4.69 | −0.032 | −0.773 | 2.907 | 0.004 |
| | Adolescents | 215 | 13.97 | 4.11 | 0.305 | −0.494 | | |
| Nurturing Parent | Parents | 215 | 20.31 | 3.30 | −0.752 | 0.200 | 3.822 | 0.000 |
| | Adolescents | 215 | 19.05 | 3.51 | −0.856 | 0.770 | | |
| Adult | Parents | 215 | 20,73 | 3.34 | −0.812 | 0.399 | 6.880 | 0.000 |
| | Adolescents | 215 | 18.47 | 3.44 | −0.462 | −0.260 | | |
| Adapted Child | Parents | 215 | 17.47 | 3.98 | −0.433 | −0.116 | 5.540 | 0.000 |
| | Adolescents | 215 | 15.37 | 3.87 | −0.032 | −0.601 | | |
| Natural Child | Parents | 215 | 18.31 | 3.43 | −0.514 | −0.783 | −1.114 | 0.266 |
| | Adolescents | 215 | 18.67 | 3.24 | −0.501 | −0.058 | | |

### 3.2. Descriptive Statistics of Life Position

Next, the results of descriptive statistics of life positions are presented in Table 2 and the values show that the leading life position in adolescents and parents is the life position "I am OK—you are OK", and the least manifested in both groups of subjects is the position "I am not OK—you are not OK" The first hypothesis was confirmed in the part about the leading role of the life position "I am OK—you are OK". The results are comparable to those of our other studies investigating life positions in adolescents and adults [24,43].

**Table 2.** Mean, standard deviation, normality indices, and differences in life positions between the adolescents and their parents.

| Dimensions | Respondents | N | Mean | SD | Skewness | Kurtosis | t-Test | p |
|---|---|---|---|---|---|---|---|---|
| "I am OK—you are OK" | Parents | 183 | 24.98 | 3.56 | −1.313 | 0.917 | 5.108 | 0.000 |
| | Adolescents | 199 | 23.10 | 3.59 | −0.675 | 0.200 | | |
| "I am OK—you are not OK" | Parents | 183 | 15.35 | 4.40 | 0.180 | −0.367 | 1.507 | 0.133 |
| | Adolescents | 199 | 14.70 | 3.98 | 0.150 | −0.475 | | |
| "I am not OK—you are OK" | Parents | 183 | 16.58 | 3.97 | 0.294 | 0.457 | −0.690 | 0.491 |
| | Adolescents | 199 | 16.85 | 3.78 | 0.326 | −0.063 | | |
| "I am not OK—you are not OK" | Parents | 183 | 12.42 | 4.39 | 0.805 | 0.711 | −2.727 | 0.007 |
| | Adolescents | 199 | 13.67 | 4.57 | 0.718 | 0.366 | | |

Life Positions Differences between the Adolescents and Their Parents

From the t-test values presented in Table 2, it is evident that there are significant differences between the life position "I am OK—you are OK", which is more pronounced in parents, and the life position "I am not OK—you are not OK", which is more pronounced in adolescents. No significant difference was found for the other two life positions. The results obtained confirm the second hypothesis put forward in the part of the existence of differences between the life positions of adolescents and their parents.

### 3.3. Descriptive Statistics of Stroke Economy

The results of the stroke economy descriptive statistics presented in Table 3 show that no one or two clear leading patterns emerged in the individuals studied. It can be said that adolescents and their parents favor the "Don't ask" stroke economy, while the "Don't reject" stroke economy is least favored by both groups of respondents. The first hypothesis put forward about the leading role of the "Don't ask" stroke economy in adolescents and their

parents was partially confirmed. Similar results have been obtained in our other studies on stroke economy in adolescents and adults in school settings [45].

**Table 3.** Mean, standard deviation, normality indices, and differences in stroke economy between the adolescents and their parents.

| Dimensions | Respondents | N | Mean | SD | Skewness | Kurtosis | *t*-Test | *p* |
|---|---|---|---|---|---|---|---|---|
| Don't Give | Parents | 213 | 10.17 | 2.67 | −0.133 | −0.585 | 3.904 | 0.000 |
| | Adolescents | 215 | 9.19 | 2.54 | 0.195 | −0.413 | | |
| Don't Ask | Parents | 213 | 10.54 | 2.69 | −0.133 | −0.551 | 1.078 | 0.282 |
| | Adolescents | 215 | 10.27 | 2.48 | −0.194 | −0.310 | | |
| Don't Accept | Parents | 213 | 10.16 | 2.57 | −0.100 | −0.527 | 2.675 | 0.008 |
| | Adolescents | 215 | 9.50 | 2.55 | −0.386 | 0.087 | | |
| Don't Reject | Parents | 213 | 8.93 | 2.86 | 0.089 | −0.419 | 1.376 | 0.170 |
| | Adolescents | 215 | 8.57 | 2.61 | 0.041 | −0.255 | | |
| Don't Give Yourself | Parents | 213 | 10.03 | 2.85 | −0.163 | −0.439 | 0.253 | 0.800 |
| | Adolescents | 215 | 9.96 | 2.48 | −0.316 | −0.152 | | |

Stroke Economy's Differences between the Adolescents and Their Parents

The *t*-test values presented in Table 3 indicate the presence of significant differences between the economy of "Don't Give" and "Don't Accept" signs of acknowledgement, which are more pronounced in parents compared to adolescents. No differences were found between the "Don't Ask", "Don't Reject" and "Don't Give Yourself" styles of stroke economy. The results confirmed the second hypothesis put forward that there are differences between the stroke economy in adolescents and their parents.

*3.4. Results of Linear Regression Analyses*

Stroke economy dimensions of adolescents were regressed on ego states and life positions. The Condition Index of ego states was less than 15 and didn't show problems of Collinearity [47]. The Condition Index of life positions was greater than 15 and may indicate a problem and for that reason they are excluded and not presented in the regression analysis results on Table 4.

The ego states explained the 15% of variance of the Don't Give Yourself stroke economy (F = 4.859; $p < 0.001$) and the best predictor was the Adult ego state (β = 0.255, t = 3.086; $p < 0.01$). Next, the ego states explained the 15% of variance of the Don't Give stroke economy (F = 4.657; $p < 0.001$) and the significant predictor was only the Normative Parent ego state (β = 0.219, t = 2.629; $p < 0.01$). The ego states explained the 13% of variance of the Don't Reject stroke economy (F = 4.067; $p < 0.001$) and the significant predictor was only the Natural Child ego state (β = 0.211, t = 2.631; $p < 0.01$). The ego states explained the 9% of variance of the Don't Ask stroke economy (F = 2.977; $p < 0.01$) and significant predictor was only the Adapted Child ego state (β = 0.200, t = 2.320; $p < 0.05$). Finally, the ego states explained the 7% of variance of the Don't Accept stroke economy (F = 2.691; $p < 0.01$) and the significant predictor was only the Nurturing Parent ego state (β = 0.184, t = 1.933; $p < 0.05$).

Stroke economy dimensions of parents were regressed on ego states and life positions. The Condition Index of ego states was <15 and did not show problems of Collinearity [47]. The Condition Index of life positions was greater than 15 and may indicate a problem and for that reason they are excluded and not presented in the regression analysis results on Table 5.

**Table 4.** Regression analyses of ego states and life positions with stroke economy dimensions as dependent variables in adolescents.

| Dimensions | Don't Give Yourself | | Don't Give | | Don't Reject | | Don't Ask | | Don't Accept | |
|---|---|---|---|---|---|---|---|---|---|---|
| | β | t | β | t | β | t | β | t | β | t |
| Normative Parent | 0.023 | 0.274 | 0.219 ** | 2.629 | −0.033 | −0.393 | −0.053 | −0.618 | 0.005 | 0.061 |
| Nurturing Parent | −0.106 | −1.166 | 0.023 | 0.247 | −0.006 | −0.065 | 0.148 | 1.566 | 0.184 * | 1.933 |
| Adult | 0.255 ** | 3.086 | 0.134 | 1.612 | −0.060 | −0.710 | 0.005 | 0.059 | 0.092 | 1.063 |
| Adapted Child | 0.061 | 0.733 | 0.162 | 1.944 | 0.087 | 1.037 | 0.200 * | 2.320 | 0.007 | 0.084 |
| Natural Child | 0.192 ** | 2.451 | 0.045 | 0.573 | 0.211 ** | 2.631 | 0.041 | 0.499 | 0.132 | 1.609 |
| *p* | *p* < 0.001 | | *p* < 0.001 | | *p* < 0.001 | | *p* < 0.01 | | *p* ≤ 0.01 | |
| $R^2$ | 0.19 | | 0.19 | | 0.17 | | 0.13 | | 0.12 | |
| $AR^2$ | 0.15 | | 0.15 | | 0.13 | | 0.09 | | 0.07 | |
| F | 4.859 | | 4.657 | | 4.067 | | 2.977 | | 2.691 | |

Note: *p* = Significance; $R^2$ = R Square; $AR^2$ = Adjusted R Square; ** $p \leq 0.01$, * $p \leq 0.05$.

**Table 5.** Regression analyses of ego states and life positions with stroke economy dimensions as dependent variables in parents.

| Dimensions | Don't Ask | | Don't Reject | | Don't Accept | | Don't Give | | Don't Give Yourself | |
|---|---|---|---|---|---|---|---|---|---|---|
| | β | t | β | t | β | t | β | t | β | t |
| Normative Parent | −0.209 * | −2.295 | 0.028 | 0.309 | −0.147 | −1.581 | 0.108 | 1.161 | −0.037 | −0.388 |
| Nurturing Parent | 0.072 | 0.749 | 0.151 | 1.559 | 0.183 | 1.870 | 0.112 | 1.145 | 0.027 | 0.269 |
| Adult | 0.081 | 0.945 | −0.018 | −0.204 | 0.025 | 0.278 | 0.011 | 0.130 | −0.043 | −0.472 |
| Adapted Child | 0.306 *** | 3.310 | 0.147 | 1.571 | 0.199 * | 2.098 | 0.087 | 0.914 | 0.189 | 1.933 |
| Natural Child | −0.044 | −0.516 | 0.216 * | 2.500 | −0.039 | −0.444 | −0.096 | −1.094 | −0.052 | −0.573 |
| *p* | *p* ≤ 0.001 | | *p* ≤ 0.01 | | *p* ≤ 0.01 | | *p* = 0.01 | | *p* > 0.05 | |
| $R^2$ | 0.16 | | 0.14 | | 0.12 | | 0.12 | | 0.06 | |
| $AR^2$ | 0.12 | | 0.09 | | 0.07 | | 0.07 | | 0.01 | |
| F | 3.643 | | 3.059 | | 2.605 | | 2.505 | | 1.244 | |

Note: *p* = Significance; $R^2$ = R Square; $AR^2$ = Adjusted R Square. *** $p \leq 0.001$, * $p \leq 0.05$.

The ego states explained the 12% of variance of the Don't Ask stroke economy (F = 3.643; $p < 0.001$) and the best predictor was the Adapted Child ego state (β = 0.306, t = 3,310; $p < 0.001$). Next, the ego explained the 9% of variance of the Don't Reject stroke economy (F = 3.059; $p < 0.01$) and the significant predictor was only the Natural Child ego state (β = 0.216, t = 2.500; $p < 0.05$). The ego states explained the 7% of variance of the Don't Accept stroke economy (F = 2.605; $p < 0.01$) and significant predictor was only the Adapted Child ego state (β = 0.199, t = 2.098; $p < 0.05$).

The results obtained and presented from the statistical analyses give us reason to summarize that there are significant differences only in the ego states as determinants of the stroke economy in adolescents and their parents, and thus partly confirm the third hypothesis put forward.

## 4. Discussion

The purpose of the present study was to examine functional ego states, life position, and stroke economy, and to disclose patterns of social interaction in adolescents and their parents. Several authors presenting cases from clinical practice indicate the presence of a connection between negative coping strategies, psychological problems, emotional

suffering and highly manifested Normative Parent and Adapted Child, as well as the life position "I am not OK" [17,28,49]. At the same time, closeness and good relationships in the family are supported by the giving of positive signs of recognition and lack of stroke economy [26,28]. According to previous studies, parents' behaviors may help adolescents to become more mature in their behavior and have effective communication, but adolescents' responsible actions also affect the degree to which parents continue to grant increasing independence, as appropriate [6].

We found that for adolescents, the Nurturing Parent, Natural Child, Adult ego states, and the life position "I am OK—you are OK" were leading. In summary, adolescents in the present study demonstrate the presence of characteristics such as empathy, caring, friendliness and rationality. They also exhibit more pronounced impulsivity, emotionality, assertive behavior, self-respect, and acceptance of others as they are. In their social interactions, they exhibit modesty, a tendency to reject deserved praise, lack confidence in their abilities, and underestimate the importance of achieving personal success. Similar results were reported in a study of the impact of transactional analysis training programs. It found that adolescents who participated in the training program demonstrated better problem-solving strategies, used the Adult ego state more effectively, and expressed more positive feelings compared to their peers [49].

According to the results obtained, the Nurturing Parent, Adult ego states and the life position "I am OK—you are OK" dominate in parents. In this regard, it can be said that parents in the present study demonstrate characteristics such as analyticity, resourcefulness, compassion, caring and responsibility. They also show assertive behavior, a tendency to follow rules and set clear boundaries in their relationships. In social interactions, parents are more likely not to accept praise and signs of appreciation, and not to seek feedback on their behavior, even when they need it. We have obtained similar results in our other studies of adults in Bulgarian contexts [45].

As we would expect, adolescents reported a greater tendency to express insecurity, negative attitudes towards themselves and others, and doubt in their coping skills and others' abilities to understand and help. We assume that this result can be related to the dynamic changes in adolescents in the period of transition from childhood to adulthood. This process is psychosocial in nature and is often related to solidarity or comparison with others in the social environment [4,6]. At the beginning of this stage of individual development, although dependent on parental support, the adolescent increasingly begins to focus on self-discovery, identity formation, independence, and preparation for leaving the family. One of the important tasks during this complex process is the unification of separate ego states into one coherent functional model [20]. Parents in the present study demonstrated greater self-confidence and self-affirming behavior, positive attitudes in relationships with others, resilience, and flexibility in difficult situations. Moreover, compared to adolescents, parents are rated as more organized, structured, analytical and rational, following generally accepted rules and accommodating the demands of others. Based on the results, it can be assumed that in their social interactions, parents are more likely to express a tendency not to give signs of recognition when others deserve them and not to accept criticism or positive feedback. These results are identical to the results obtained in our other studies of adolescents and adults in the Bulgarian context [27,45]. This gives us reason to assume that the parents in the present study form a social interaction behavior pattern that is more characterized by the manifestation of restraint, criticality and stroke economy. This result may be due both to the specifics of the parents who participated in the study, and to the cultural characteristics of the parenting style.

The weaker expression of the Normative Parent ego state and the "I am not OK—you are not OK" life position in adolescents and parents suggests that they demonstrate more flexibility in terms of following the rules, exhibit less critical and rigidity in their behavior, have developed a positive self-esteem, while also expressing a positive opinion of others. The results also indicate that parents and adolescents do not differ in seeking and refusing positive or negative signs of recognition in their social interactions with others. The results

also suggest that parents and adolescents do not differ in seeking and rejecting positive or negative signs of recognition in their social relationships with others. Similar results were obtained in our other studies on signs of recognition in adolescents and adults in a school environment [45], but according to the results of other authors, the content of the messages, as well as the kind of social interaction of adolescents and parents, differ [3,50]. These contradictory results may be due to various factors. For example, close family relationships have been found to influence adolescent development and adaptation [51,52]. Next, it can be said that continuity exists in parent-child relationships, which sometimes continues across the life span [50,53]. It is also important to note that parent-adolescent relationships influence young people's adjustment to the social environment, as well as their psychosocial and psychological well-being [40,54,55].

The results of the regression analysis indicate that functional ego states have a stronger and more significant positive effect on the stroke economy in adolescents (see Table 4) than their parents (see Table 5). This suggests that characteristics such as analyticity, objective assessment of reality, commitment to problem solving, striving for success, and impulsivity lead to higher self-criticism, underestimation of one's own needs, and a decreased experience of personal satisfaction in adolescents. Next, it can be concluded that criticality, imposing one's own opinion, judgmental and evaluative behavior lead to lack of praise and encouragement, as well as less giving of strokes. In summary, spontaneity, emotionality, positive self-esteem, and striving to satisfy primarily one's own needs weakly increase the ability in adolescents to accept any signs of recognition and not to refuse them, whether they are deserved or not. This result suggests that conforming, over adaptive and submissive behaviors in the subjects lead to increased difficulty in asking for help, support, or feedback from others. In general, empathic, caring, and supportive adolescents are more likely to refuse and not accept strokes related to themselves or their behaviors.

In summary, the present study found two different patterns of influence on social interaction patterns in adolescents and their parents. In adolescents, each of the five ego states have a significant effect on only one of the stroke economy patterns, contributing to its higher expression. Specifically, adolescents with rational, self-confident, and risk-taking behaviors tend to underestimate their own needs and achievements, and not give themselves signs of recognition to a greater extent. In addition, critical behavior, restraint and rigidity are a prerequisite for adolescents not to appreciate others; it is difficult for them to give positive signs of recognition or constructive suggestions. Next, it can be assumed that spontaneity, impulsivity and the pursuit of pleasure increase the tendency of adolescents not to reject undeserved criticism and praise. The results give us reason to assume that adaptation to the demands of others, the desire to be liked, and the tendency to obey increase their anxiety and feelings of inferiority, and lead to the inability to seek help when they need it.

In contrast to the case of the adolescents, only three functional ego states had a significant effect in determining of stroke economy in the case of the parents. According to the results, characteristics such as consideration for others, efficiency, and conformity are prerequisites for parents to seek recognition for their behavior, as well as to reject received praise or criticism, regardless of whether it is deserved or not. In summary, the manifestations of control and self-control, high demands on others and a critical attitude also imply the search for feedback and verification of the correctness of their behavior in parents studied. We can conclude that characteristics such as sociability, emotionality, spontaneity and friendliness lead to acceptance of the signs of recognition from others in the process of parents' social interaction. Similar conclusions are drawn by other authors, according to whom the behavior in social relationships of parents differs from that of adolescents [50] and is influenced by various factors such as the degree of social maturity, the ability to positively or negatively exchange with others and cognitive and emotional response to circumstances [3].

The Nurturing Parent and Adult ego states were found not to be significant predictors of any dimensions of the stroke economy in the parents studied. This may be due on the one

hand to the assumed greater maturity, social competence and autonomy of the parents, and on the other hand may be due to the specific characteristics of the personality constructs studied. The Nurturing Parent and Adult ego states imply the manifestation of a positive attitude towards others, and the presence of better expressed social and communicative skills, which is related to the exchange of many positive strokes in the process of social interaction [1,18,26,28]. Other authors also assume that precisely because of their greater experience and skills, parents can be an important model for effective interaction and the adjustment of adolescents to the social environment during the period of transition [41].

*Strengths and Limitations of the Study*

The relatively large sample size, the study of adolescents and their parents and the use of questionnaires with good psychometric values can be considered the strengths of this study. The use of quantitative methods for the study of personality dimensions is also, in our opinion, a strength, which can add to the knowledge on a transactional analysis and developmental psychology in general in terms of knowledge on the functional ego states, life positions and stroke economy of adolescents and their parents. Limitations of the study are related to the lack of similar quantitative studies available to the authors, which does not allow the conclusions obtained for adolescents and parents to be compared with results from other similar studies in the field of transactional analysis. It is also possible that the specificity of the sample, the period of youth transition, and the social context affect the results of adolescents. Another limitation of the study is the use of self-report questionnaires, which may be influenced by social desirability. In addition, interactions between factors at different levels (social, interpersonal, individual) that have not been investigated may also have an effect on the results obtained.

## 5. Conclusions

The results obtained in the present study give us grounds to conclude that ego states have a significant effect on the stroke economy and exchange of signs of recognition in social relationships of adolescents and their parents.

According to the results, the five functional ego states positively influence the stroke economy in adolescents. At the same time, in parents, the ego states Normative Parent, Adapted Child, and Natural Child influence the stroke economy. Based on the obtained results, we can summarize that at the age of transition from childhood to adulthood, adolescents revise and update of the patterns of interaction that they formed in early childhood. Ego states were found to have a positive effect on the stroke economy, indicating that adolescents are becoming more flexible and beginning to connect their social relationships with their inner subjective experiences. At the same time, it can be said that in this important period of transition, parents and other significant adults can be good role models and examples of interaction with the social environment. Therefore, it can be concluded that by manifesting assertive behavior, setting clear boundaries and being selective in expressing and accepting signs of recognition, parents can teach the adolescents to have effective patterns of behavior and relationships with others.

We hope that this study will stimulate other research on the relationships between ego states, life positions, strokes and other psychological phenomena in a family environment (e.g., adolescent-parent communication styles, well-being, coping strategies, life satisfaction, etc.). Results of such studies will lead to the accumulation of new empirical data and at the same time would be useful for professionals practicing in various fields such as psychotherapy, education, counseling and personal development. The results presented in this article study can be applied by and useful to psychologists, teachers and other helping professionals in counseling parents and adolescents to improve their relationships, manage conflict, and prevent risky behaviors.

**Author Contributions:** Conceptualization, S.B. and E.T.; methodology, S.B.; validation, E.T.; formal analysis, S.B. and E.T.; investigation, S.B.; resources, S.B. and E.T.; data curation, S.B.; writing—original draft preparation, S.B.; writing—review and editing, E.T.; visualization, S.B.; supervision, E.T.; project administration, S.B.; funding acquisition, E.T. All authors have read and agreed to the published version of the manuscript.

**Funding:** This research received no external funding. The data collection for the study was funded at the national level.

**Institutional Review Board Statement:** This study was conducted according to ethical principles and approved by the Ministry of Education and Science, Bulgaria (protocol code: N 09-118 and date of approval 14.06.2019).

**Informed Consent Statement:** Informed consent was obtained from all subjects involved in the study. Parents also provided informed consent for adolescents' participation and adolescents provided assent for participation. Participation was voluntary and confidential. The article presents neither individual- nor school-level data.

**Data Availability Statement:** The data presented in this study are available on reasonable request from the authors.

**Acknowledgments:** This work was supported by the Bulgarian Ministry of Education and Science under the National Research Programme "Young scientists and postdoctoral students" approved by DCM # 577/17.08.2018.

**Conflicts of Interest:** The authors declare no conflict of interest.

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
