# Peer review of "Functional Ego States, Behavior Patterns, and Social Interaction of Bulgarian Adolescents and Their Parents"

_societies, doi:10.3390/soc13070154_

Round 1

Author Response

Cover letter on the suggestions from Reviewer 1

I want to thank you for the opportunity to review this manuscript. The time spent creating and submitting it is greatly appreciated.

We are grateful to the reviewer for making correct, important, and detailed comments and suggestions that helped us to improve our manuscript.

We will respond briefly to each comment. We accept all comments and suggestions and have made the necessary edits to the text of the manuscript.

In our responses to each comment, we have referenced in which line the changes in the manuscript can be checked.

However, I consider that, despite the fact that the topic is interesting, the following changes are necessary for its possible publication:

Introduction: Authors write: “The role of the family in the process of individual development is perhaps the most researched topic by authors studying adolescence”. Please, clarify better studies that analyze this aspect. Lines 25-30; lines 76-78: Which are studies?

The sentences mentioned have been edited, and additional sources have been added to them. The changes are in Lines 28-31 and Lines 93-102.

Method: Please, include lines 158-177 at the end of the Introduction.

Thank you for the recommendation! Text edited and moved at the end of the section Introduction Lines 58-61, както и the objective and hypotheses have been displaced in Lines 208 – 227, before the  Method section.

Please, explain better the link between the literature and the rationale of this study.

Adolescence is a dynamic period of transition from childhood to adulthood. This is the time when the adolescent has several challenges, such as integrating his early experiences, building an adequate self-image, and forming effective relationships. The family is the social environment in which the quality of the relationship with parents is a significant factor in the development of self-esteem, a sense of self-worth, and self-control. Although it is well known that parents’ ways of raising their children significantly affect their children’s personality and behavior, this study has been designed because there is a lack of specific research on which ego states and life positions are associated with stroke economy in adolescents and their parents.

We try to give an additional explanation of the link between the literature and the rationale of this study in the manuscript -  Lines 51-67 and Lines 184-207.

Where is the approval of the ethics committee?

Thank you for the questions!

The approval of the ethics committee has been added in Lines 575-577.

Please, write examples of items for each instrument

The subsection introducing the research tools has been edited. We added the recommended additional information in Lines 246-277.

Please, write the descriptive statistics section, in which data screening shall be reported.

The statistical analysis methods used in the study have been supplemented and presented as a separate subsection in the Methods section, Lines 281-293.

In tables 1, 2, and 3 skewness and kurtosis normality indices have been added.

Results: Regression analysis is very messy. Please, see the following paper: “The Role of Gender in the Association Among the Emotional Intelligence, Anxiety and Depression” by Sergi et al. to study the regression analysis.

Thank you for the recommendations made and the proposed example for presenting the results of the regression analyses.

Corresponding corrections have been made in Tables 4 and 5 at Lines 347-383.

Regression analysis is explained in a suggested way and additional information about t and AR2 values have been added.

The discussion resembles a discussion of the research results, not a comparison of the results of your own research with other authors. What is the significance of these findings in relation to the wider body of the literature? What are the limitations and and the implications of this study?

Thank you for the objective directions. We mainly edited the Discussion section, trying to compare our results with other authors' research. We have outlined the main findings of our study and the relationship of these results to the wider literature - Lines 402-525.

A new subsection on strengths and limitations has been added - Lines 526-540.

Possible applications of the results are indicated in Lines 563-566.

In the conclusions part of the study, suggestion or suggestions for future studies can be given

Thanks for the comment! We have added suggestions for future studies on Lines 556-563.

Reviewer 2 Report

Firstly, the research is generally well constructed and very interesting. Congratulations to the authors for their work.

The theme studied, adolescence and interaction with parents as an important element to go through it successfully, is interesting and necessary. Specifically, this paper show results from a study of functional ego states, life positions, and stroke economy in adolescents and their parents as personality constructs proposed in transactional analysis theory. An assessment of each part of the brief is discussed below. Finally, an overall opinion of the paper is presented.

1. The abstract is well structured, allowing the reader to know the objective of the research, the method used and the main results found.

2. Introduction: A good job of contextualising the study is done. In this way, different parts of the study are addressed, which are necessary to understand in order to understand the work being carried out. It is a coherent introduction/analytical framework with a correct bibliography.

I think that the last paragraph of this section that begins with " It can be summarized that in each family a certain pattern of interaction and behavior 149 develops,…" is very clarifying and is very well done and placed.

3. Methods: It begins by clearly and unambiguously stating the objective of the research. From these, tasks (specific objectives) and hypotheses are constructed and presented. There is coherence between objectives and hypotheses.

As for the sample, I have some doubts: what kind of sampling has been carried out? why? I think this is a fundamental aspect for a research to be published, as it is what makes it reproducible. I believe that these aspects should be included in this section.

It would also be interesting to discuss in this section the type of data analysis leading to the results. It should be explained why a linear regression and not another type.

4. Results: The results found are clearly shown and are interesting. They are well structured, which allows for a good understanding. Below each table I would add a legend: source: own elaboration (these are results and tables elaborated by the authors themselves).

5. Conclusions: The conclusions reached are interesting and in line with the results found and the literature reviewed.

However, I think it is necessary to highlight two aspects: on the one hand the limitations of this study and on the other hand the future lines of research that emerge from it.

Author Response

Cover letter on the suggestions from Reviewer 2

Firstly, the research is generally well constructed and very interesting. Congratulations to the authors for their work.

We are grateful to the reviewer for making correct, important, and detailed comments and suggestions that helped us to improve our manuscript.

We will respond briefly to each comment. We accept all comments and suggestions and have made the necessary edits to the text of the manuscript.

In our responses to each comment, we have referenced in which line the changes in the manuscript can be checked.

The theme studied, adolescence and interaction with parents as an important element to go through it successfully, is interesting and necessary. Specifically, this paper show results from a study of functional ego states, life positions, and stroke economy in adolescents and their parents as personality constructs proposed in transactional analysis theory. An assessment of each part of the brief is discussed below. Finally, an overall opinion of the paper is presented.

Point 1: The abstract is well structured, allowing the reader to know the objective of the research, the method used and the main results found.

Response 1: Thank you for your comment and positive evaluation of our work!

Point 2: Introduction: A good job of contextualising the study is done. In this way, different parts of the study are addressed, which are necessary to understand in order to understand the work being carried out. It is a coherent introduction/analytical framework with a correct bibliography.

I think that the last paragraph of this section that begins with " It can be summarized that in each family a certain pattern of interaction and behavior 149 develops,…" is very clarifying and is very well done and placed.

Response 2: Thank you for your comment and positive evaluation of our work!

Point 3: Methods: It begins by clearly and unambiguously stating the objective of the research. From these, tasks (specific objectives) and hypotheses are constructed and presented. There is coherence between objectives and hypotheses.

As for the sample, I have some doubts: what kind of sampling has been carried out? why? I think this is a fundamental aspect for a research to be published, as it is what makes it reproducible. I believe that these aspects should be included in this section.

Response 3: Thank you for the recommendations made and the questions asked.

A more detailed description of the sample was made and is presented in the edited manuscript on Lines 228-244.

Point 4: It would also be interesting to discuss in this section the type of data analysis leading to the results. It should be explained why a linear regression and not another type.

Response 4: Thank you for the objective directions. The statistical analysis methods used in the study have been supplemented and presented as a separate subsection in the Methods section, Lines 281-293.

Because the study aimed to test the predictive role of ego states and life positions on the stroke economy, linear regression analysis was preferred as the statistical analysis method. We gave an additional explanation of the link between the literature and the rationale of this study in the manuscript -  Lines 51-67 and Lines 183-206.

Point 5: Results: The results found are clearly shown and are interesting. They are well structured, which allows for a good understanding. Below each table I would add a legend: source: own elaboration (these are results and tables elaborated by the authors themselves).

Response 5: Thank you for your comment!

We added the text „Tables and results shown on them are elaborated by the authors themselves“ in the subsection Statistical Analysis. The aim was to avoid repetition under each table - Lines 292-293.

Point 6: Conclusions: The conclusions reached are interesting and in line with the results found and the literature reviewed.

However, I think it is necessary to highlight two aspects: on the one hand the limitations of this study and on the other hand the future lines of research that emerge from it.

Response 6: Thank you for the comments!

A new subsection on strengths and limitations has been added - Lines 526-539. The author’s suggestions for future studies have been added - to Lines 557-562.

Reviewer 3 Report

I have read this article several times in order to figure out how I can be helpful to the authors. I am very surprised that the authors chose the theoretical orientation of transactional analysis (TA). This was developed in the 1950s by Eric Berne and expounded upon by Claude Steiner. Both Berne and Steiner deserve to be named in this article and TA described in some detail. The authors then need to provide a reasonable explanation for why TA is especially relevant to today's world and not simply an important part of history.

As indicated above, the literature review is poor, with mostly older references. If I'm not mistaken, the most recent reference regarding TA is from 2002.

The study sample is recent and is a good size--215 adolescents and 215 parents. In line # 425 the authors state the size of the sample is a limitation. I don't agree. Another limitation is described as few quantitative studies incorporating TA (line #s 425-426), evidence that the relevance is not what it was considered almost 70 years ago. Line #s 415-417 state that there are no studies about TA constructs as they apply to adolescents and parents. Thus, the authors do not seem to be aware that these constructs are no longer utilized.

There is a good description of the two self-assessment questionnaires and the self-report questionnaire.

I am not a research methodologist/statistician and I hope another reviewer can be helpful here.

Line #437 "learn" should be "teach"

A strength of this study is that it points to the importance of healthy family relationships and that parenting is an exceptionally important factor in the development of children and adolescents. We learn that adolescents have greater insecurity, negative attitudes toward themselves and others, doubt their own coping skills and are hesitant to reach out for help.

Overall, the authors have put in a great deal of work on this article. I hope they are successful in making the compelling case for the use of TA.

English language is good.

Author Response

Cover letter on the suggestions from Reviewer 3

I have read this article several times in order to figure out how I can be helpful to the authors.

We are grateful to the reviewer for making correct, important, and detailed comments and suggestions that helped us to improve our manuscript.

We will respond briefly to each comment. We accept all comments and suggestions and have made the necessary edits to the text of the manuscript.

In our responses to each comment, we have referenced in which line the changes in the manuscript can be checked.

Point 1: I am very surprised that the authors chose the theoretical orientation of transactional analysis (TA). This was developed in the 1950s by Eric Berne and expounded upon by Claude Steiner. Both Berne and Steiner deserve to be named in this article and TA described in some detail. The authors then need to provide a reasonable explanation for why TA is especially relevant to today's world and not simply an important part of history.

Response 1: We agree that Eric Berne and Claude Steiner should be referenced and presented with dignity.  We understand that we have only briefly mentioned their enormous contribution to TA, which does not mean that we have underestimated both authors. We added additional information in Lines 69-71 and Lines 120-126. Although it was created 70 years ago, TA continues to inspire many authors, and several books in recent years have improved Berne's theory. Nowadays the International TA Association (ITAA) and European Association for TA (EATA) have thousands of members who apply their knowledge and skills in four main areas: psychotherapy, education, organizations, and counseling.

Point 2: As indicated above, the literature review is poor, with mostly older references. If I'm not mistaken, the most recent reference regarding TA is from 2002.

Response 2: An additional literature review was conducted and new TA articles were examined and published primarily in the Journal of Transactional Analysis. 17 new sources have been added, of which 3 are about statistical analyses, 6 are in the field of TA, and 8 are related to adolescent-parent relationships.

Point 3: The study sample is recent and is a good size--215 adolescents and 215 parents. In line # 425 the authors state the size of the sample is a limitation. I don't agree. Another limitation is described as few quantitative studies incorporating TA (line #s 425-426), evidence that the relevance is not what it was considered almost 70 years ago. Line #s 415-417 state that there are no studies about TA constructs as they apply to adolescents and parents. Thus, the authors do not seem to be aware that these constructs are no longer utilized.

Response 3: A new subsection on strengths and limitations has been added - Lines 526-539.

Although few in number, up-to-date research in the field of TA (Lines 93-102) and the supposed reasons for the lack of quantitative research in TA (Lines 51-61) were indicated.

Point 4: There is a good description of the two self-assessment questionnaires and the self-report questionnaire.

Response 4: Thank you for the comment.

Additional information about the questionnaires has been added - Lines 246-276.

Point 5: I am not a research methodologist/statistician and I hope another reviewer can be helpful here.

Response 5: Presented results of the statistical analysis were edited and improved according to the recommendations of the other reviewers -  Lines 281-293.

Point 6: Line #437 "learn" should be "teach"

Response 6: We accept the recommendation. We edited the sentence - Line 555.

Point 7: A strength of this study is that it points to the importance of healthy family relationships and that parenting is an exceptionally important factor in the development of children and adolescents. We learn that adolescents have greater insecurity, negative attitudes toward themselves and others, doubt their own coping skills and are hesitant to reach out for help.

Overall, the authors have put in a great deal of work on this article. I hope they are successful in making the compelling case for the use of TA.

Response 7: Thank you for the positive evaluation of our work. We know the difficulty of quantitatively examining the psychodynamic TA constructs. We have motivated and pointed out arguments for our choice, specifically in Lines 51-61 and Lines 184-207.

After editing the manuscript, it was submitted to a colleague for extensive English revisions and the appropriate improvements were made to the text.

Round 2

Reviewer 1 Report

Authors followed the proposed revisions.

Reviewer 3 Report

This is a significantly improved manuscript! I applaud the authors for making the case of the relevance of TA today and for adding to the reference list. It is evident that the authors took reviewers' comments seriously and I commend them for that.

There are still a few typos in the manuscript.